# Gain-of-function cardiomyopathic mutations in RBM20 rewire splicing regulation and re-distribute ribonucleoprotein granules within processing bodies

Aidan M. Fenix[1,2,3,15], Yuichiro Miyaoka[4,5,15], Alessandro Bertero[1,2,3], Steven M. Blue[6], Matthew J. Spindler[5], Kenneth K. B. Tan[5], Juan A. Perez-Bermejo[5], Amanda H. Chan[5], Steven J. Mayerl[5], Trieu D. Nguyen[5], Caitlin R. Russell[5], Paweena P. Lizarraga[5], Annie Truong[5], Po-Lin So[5], Aishwarya Kulkarni[7,8], Kashish Chetal[8], Shashank Sathe[6], Nathan J. Sniadecki[1,2,3,9,10], Gene W. Yeo[6], Charles E. Murry[1,2,3,10,11,12✉], Bruce R. Conklin[5,13✉] & Nathan Salomonis[8,14✉]

Mutations in the cardiac splicing factor RBM20 lead to malignant dilated cardiomyopathy (DCM). To understand the mechanism of RBM20-associated DCM, we engineered isogenic iPSCs with DCM-associated missense mutations in RBM20 as well as RBM20 knockout (KO) iPSCs. iPSC-derived engineered heart tissues made from these cell lines recapitulate contractile dysfunction of RBM20-associated DCM and reveal greater dysfunction with missense mutations than KO. Analysis of RBM20 RNA binding by eCLIP reveals a gain-of-function preference of mutant RBM20 for 3′ UTR sequences that are shared with amyotrophic lateral sclerosis (ALS) and processing-body associated RNA binding proteins (FUS, DDX6). Deep RNA sequencing reveals that the RBM20 R636S mutant has unique gene, splicing, polyadenylation and circular RNA defects that differ from RBM20 KO. Super-resolution microscopy verifies that mutant RBM20 maintains very limited nuclear localization potential; rather, the mutant protein associates with cytoplasmic processing bodies (DDX6) under basal conditions, and with stress granules (G3BP1) following acute stress. Taken together, our results highlight a pathogenic mechanism in cardiac disease through splicing-dependent and -independent pathways.

[1] Department of Laboratory Medicine and Pathology, University of Washington, 1959 NE Pacific Street, Seattle, WA 98195, USA. [2] Center for Cardiovascular Biology, University of Washington, 850 Republican Street, Brotman Building, Seattle, WA 98109, USA. [3] Institute for Stem Cell and Regenerative Medicine, University of Washington, 850 Republican Street, Seattle, WA 98109, USA. [4] Regenerative Medicine Project, Tokyo Metropolitan Institute of Medical Science, Tokyo 156-8506, Japan. [5] Gladstone Institutes, 1650 Owens St, San Francisco, CA 94158, USA. [6] Department of Cellular and Molecular Medicine, Stem Cell Program, and Institute for Genomic Medicine, University of California San Diego, La Jolla, CA 92093, USA. [7] Department of Electrical Engineering and Computer Science, University of Cincinnati, Cincinnati, OH 45221, USA. [8] Division of Biomedical Informatics, Cincinnati Children's Hospital Medical Center, Cincinnati, OH 45229, USA. [9] Department of Mechanical Engineering, University of Washington, 3720 15th Avenue NE, Seattle, WA 98105, USA. [10] Department of Bioengineering, University of Washington, 3720 15th Avenue NE, Seattle, WA 98105, USA. [11] Department of Medicine/Cardiology, University of Washington, 1959 NE Pacific Street, Seattle, WA 98195, USA. [12] Sana Biotechnology, 188 E Blaine Street, Seattle, WA 98102, USA. [13] Department of Medicine, Cellular and Molecular Pharmacology, and Ophthalmology, University of California San Francisco, San Francisco, CA 94158, USA. [14] Department of Pediatrics, University of Cincinnati, Cincinnati, OH 45229, USA. [15] These authors contributed equally: Aidan M. Fenix, Yuichiro Miyaoka. ✉email: murry@uw.edu; bconklin@gladstone.ucsf.edu; nathan.salomonis@cchmc.org

Dilated cardiomyopathy (DCM) is the most common indication for heart transplantation[1-3]. Recent insights into the genetics of DCM have revealed over 50 DCM causal genes that involve a wide variety of cellular processes and represent high priority targets for precision therapies (Schultheiss et al.[3], McNally et al., 2013). These studies have expanded the scope of cardiomyopathy research well beyond cardiac energetics, conduction, or contractility[1,3-6]. In particular, accumulating evidence implicates defects in RNA splicing and protein quality control in heart failure pathogenesis[7-12]. Indeed, DCM can be caused by mutation or dysregulation of multiple splicing factors[13,14]. Because the altered regulation of a single splicing factor can impact broad splicing regulatory networks, the effect of splicing factor mutations is multifaceted, leading to disruption of many cardiac signaling, transcriptional, and structural pathways.

The RNA-binding motif protein 20 (RBM20) is a splicing regulator primarily expressed in heart and skeletal muscle, where it plays a central role in cardiac physiology. RBM20 directly binds to the primary RNA (pre-mRNAs) of many cardiomyopathy-associated genes where, by a process of exon exclusion, it ensures the proper production of adult protein isoforms (i.e., splice isoforms associated with cardiac maturation)[7,15-18]. Autosomal dominant mutations in RBM20 account for up to 3% of DCM cases[7,18,19]. Further, RBM20 DCM is highly penetrant, is associated with life-threatening ventricular arrhythmias, and displays earlier age of onset than DCM associated with mutations in other proteins (e.g., laminA/C or Titin). RBM20 DCM missense mutations are enriched in a stretch of five amino acids (RSRSP, which contains the R636S mutation) in the arginine–serine-rich (RS) domain of the protein[17,19]. Surprisingly, nonsense mutations or missense mutations in other portions of the 1227-amino acid-long protein are relatively rare, suggesting that they are either undiagnosed due to a mild phenotype or not tolerated[19,20].

WT RBM20 has been shown to repress exon inclusion in key regulators of cardiac excitation–contraction coupling such as TTN, CAMK2D, and RYR2, through binding to a UCUU consensus motif in adjacent introns[7]. Isolated cardiomyocytes (CMs) from rodents lacking RBM20 demonstrate prolonged action potentials and striking calcium handling defects including increased calcium transit amplitude, increased sarcoplasmic reticulum and diastolic calcium levels, and spontaneous calcium sparks[21]. While the loss of RBM20 expression leads to DCM, cardiac fibrosis, sudden death, and arrhythmias in rodent models, few studies have focused on the precise role of mutant forms of the protein that correspond to human disease[16,21]. Crucially, direct analysis of RBM20 RNA-binding sites by cross-linking immunoprecipitation (CLIP) sequencing approaches has only been performed in wild-type (WT) rodent CMs and HEK293 cells[7], thus whether mutant and WT RBM20 have distinct RNA binding preferences remains unclear.

Studies of the effect of pathogenic RBM20 mutants have spanned human, cell, and animal models. Analyses of resected human DCM hearts from individuals carrying RBM20 mutations (R636S or S635A) have identified intriguing global differences in splice-isoform and circular RNA expression, which are associated with the regulation of cardiac contractility[7,16]. CM differentiation of human-induced pluripotent stem cells (iPSCs) harboring pathological forms of RBM20 has identified calcium handling defects and splicing defects previously observed in rodent knockout (KO) models[16,21-23]. In an impressive recent effort, pigs were engineered with either heterozygous (HTZ) or homozygous (HMZ) R636S mutations in RBM20[24]. This model led to three crucial observations: (1) RBM20 R636S HMZ mutation leads to highly penetrant neonatal lethality due to heart failure; (2) mutant RBM20 co-localizes with stress granules in the CM cytoplasm after metabolic stress induced by sodium arsenate; and

(3) RBM20 undergoes an apparent liquid–liquid phase separation. However, these studies raised intriguing questions regarding the role of RBM20 in pathogenesis, physiology, normal cardiac development, and downstream RNA biology.

To address these questions, we used genome editing to generate an allelic series of RBM20 mutants, including one of the point mutations in the RS domain, R636S, as well as an RBM20 KO, in human iPSCs from a healthy subject. Normal splice patterns were differentially perturbed in RBM20 mutants as compared to KO iPSC-CMs, and genome-wide profiling of RBM20–RNA interactions revealed that mutant RBM20 showed a preference for the 3′ UTR of mRNAs. We observed co-localization of mutant RBM20 with cytoplasmic processing bodies, sites of mRNA storage and turnover, and with the canonical stress granule marker G3BP1, specifically following acute stress induction. Functional characterization of RBM20 R636S heterozygote 3D engineered heart tissues (3D-EHTs) recapitulated aspects of DCM, providing a powerful model system. Collectively, our results indicate both splicing-dependent and splicing-independent mechanisms for RBM20 DCM pathogenesis and suggest that RS domain mutant RBM20 has dominant-negative, gain-of-function properties.

## Results

### Engineering iPSC-CMs with the RBM20 R636S mutation or RBM20 KO.

To model RBM20 cardiomyopathy mutants within iPSC-CMs, we selected a commonly occurring RS domain RBM20 missense mutation: R636S (DNA: C1906A)[17]. The heterozygous mutation (R636S HTZ) was introduced into iPSCs derived from a healthy male subject (WTC11) using transcription activator-like effector nucleases (TALENs) and a droplet digital PCR (ddPCR) based strategy (Fig. 1a, b)[25,26]. To create HMZ R636S mutant iPSCs, we retargeted the WT allele of the R636S HTZ iPSC line using an allele-specific sgRNA together with dual Cas9 nickases to create an allele with the R636S mutation and a silent mutation (T1905A) that distinguishes the newly created R636S plus silent mutation (SM) allele from the original WT and R636S alleles (R636S HMZ; Fig. 1b and Supplementary Fig. S1a–c). To compare the R636S mutation with a loss-of-function mutation, we utilized our previously generated WTC11 iPSC line with an RBM20 HMZ 8-bp deletion (1917–1924del; RBM20 KO) (Fig. 1c)[25]. We confirmed that these cell lines maintained the normal male karyotype as well as loss of RBM20 protein in KO iPSC-CMs by immunofluorescent staining (Extended Data Fig. 1d–f)[25]. Importantly, we were able to generate iPSC-CMs from RBM20 mutants of high purity in all four cell lines (Supplementary Fig. 1g).

### R636S mutant iPSC-CMs exhibit electrophysiological and contractile abnormalities.

Approximately 30% and 44% of RBM20 DCM patients display conduction system disorders and malignant ventricular arrhythmias, respectively. To test whether RBM20 mutant iPSC-CMs display electrophysiological aberrations, we used multi-electrode arrays (MEAs) to record their spontaneous field potential changes (Fig. 1d and Supplementary Video 1). While we observed some biological variability across different iPSC-CM batches, the trends between biological replicates were consistent (Fig. 1d). Field potential duration corrected for beat rate differences (FPDc), an in vitro surrogate of the QT interval and biomarker for arrhythmogenesis risk[27], was not altered in RBM20 mutants. The beat rate was non-significantly reduced in RBM20 KO cells ($p = 0.09$), and RBM20 R636S HTZ iPSC-CMs displayed a statistically significant increase in spike amplitude, while RBM20 KO iPSC-CMs displayed a decrease in spike amplitude (Fig. 1d). In addition, conduction velocity was

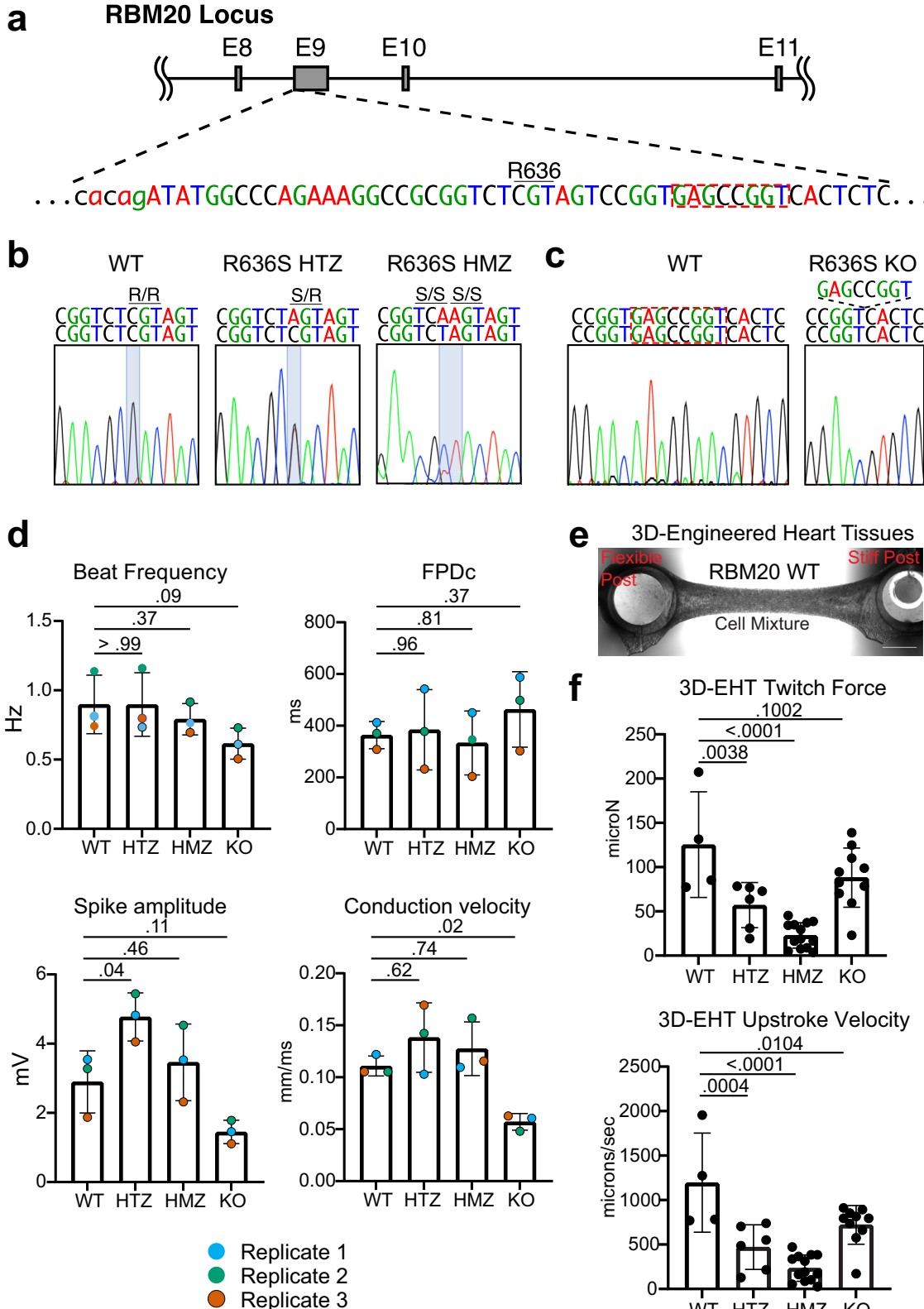

**Fig. 1 Generation and functional characterization of mutant iPSC-CMs. a** Targeted genomic region of RBM20. The location of R636 and the 8 nucleotides deleted in the 8-bp Del HMZ line are highlighted. **b** Genomic sequences of the generated RBM20 HTZ and HMZ iPSC lines. **c** Genomic sequence of the generated RBM20 8-bp Del HMZ line (KO). **d** iPSC-CM electrophysiological parameters from multi-electrode array (MEA) recordings. Color-coded paired replicates represent biologically independent experiments run in parallel. $N = 3$ biologically independent experiments. Data represented as mean values ± SEM. **e** Representative 3D-engineered heart tissues (3D-EHT) from WT and RBM20 mutant cell lines. Phase-contrast microscopy, scale bar: 1 mm. The location of flexible and glass-stiffened silicone posts is indicated. **f** Contractile metrics of 3D-EHTs. WT: $n = 4$, HTZ: $n = 6$, HMZ: $n = 13$, KO: $n = 10$. Data represented as mean values ± SEM. Statistical significance was calculated using one-way ANOVA with Dunnett's multiple comparisons test.

significantly decreased specifically in RBM20 KO iPSC-CMs (Fig. 1d).

The hallmark of RBM20 DCM is decreased cardiac contractility. To assess cardiac contractility in RBM20 mutant iPSC-CMs, we generated three-dimensional EHTs (3D-EHTs), an established model to promote iPSC-CM maturation closer to adult-like myocardium and performed more predictive disease modeling experiments (Fig. 1e and Supplementary Videos 2, 3)[28–31]. Importantly, to ensure comparability across conditions we enriched hiPSC-CMs to >90% purity through metabolic selection with sodium lactate prior to 3D-EHT casting (Supplementary Fig. 1g). RBM20 R636S HTZ and HMZ 3D-EHTs displayed significantly decreased force generation and upstroke velocity, recapitulating aspects of DCM (Fig. 1f). Intriguingly, RBM20 KO 3D-EHTs demonstrated a less dramatic decrease than the RBM20 point mutations in both these parameters and were not significantly altered compared to WT controls (Fig. 1f). These data are in line with a recent report indicating that RBM20 KO mice have normal cardiac contractility[32]. Collectively, these electrophysiology and contractility analyses demonstrate distinct functional effects of RBM20 missense and nonsense mutation, suggesting that R636S may be a gain-of-function mutation.

**RBM20 is mis-localized in R636S mutant iPSC-CMs.** To better understand the cellular mechanisms underlying the DCM-like phenotypes in R636S mutant HTZ and HMZ iPSC-CMs, we examined the intracellular localization of the mutant protein. RBM20 is known to localize to the nucleus, where it forms discrete foci that exquisitely co-localize with the site of *TTN* transcription[33]. However, several groups have demonstrated that both RSRSP deletion mutants and DCM point mutations within the RSRSP stretch, including the R636S mutation, result in mis-localization of overexpressed (exogenous) RBM20 from the nucleus to the cytoplasm[24,32,34–37]. Confocal analyses indicated that endogenous RBM20 in WT iPSC-CMs localized to the nucleus and was enriched in prominent foci (usually two per

nucleus, as immature iPSC-CMs are generally diploid), while R636S HTZ and R636S HMZ iPSC-CMs displayed more numerous and smaller RBM20 puncta which appeared primarily cytoplasmic (Fig. 2a). To assess this aspect more robustly and exclude any potential artifacts of maximum intensity image visualization of diffraction-limited microscopy, we turned to structured illumination microscopy (SIM), which affords a ~2× increase in both lateral and axial resolution compared to conventional techniques (e.g., confocal and wide-field; Fig. 2b)[38–40]. 3D axial projections of SIM data confirmed that RBM20 in R636S HTZ and HMZ iPSC-CMs was present in *both* the nucleus and cytoplasm, with a majority of RBM20 in R636S HMZ cells localized to the cytoplasm (Fig. 2b, c). These results indicate that contrary to previous reports, the exclusion of mutant RBM20 from the nucleus is not complete.

**R636S and WT RBM20 mediate distinct mRNA interactions.** To define the molecular mechanisms underlying R636S-associated defects leading to DCM, we performed an unbiased survey of direct targets for WT and mutant RBM20 binding using enhanced cross-linking immunoprecipitation sequencing (eCLIP) (Fig. 3a and Supplementary Fig. 2a)[41]. RBM20 eCLIP of WT iPSC-CMs identified 1240 reproducible peaks in biological replicates corresponding to 204 genes, including numerous well-established RBM20 splicing targets (e.g., *TTN, CAMK2D, RYR2, LDB3, LMO7, LRRFIP1, MLIP, OBSCN, SORBS2, TNN2*) (Supplementary Data 1, 2). Notably, these include prior presumed RBM20 direct targets not previously evidenced in rat heart HITS-CLIP (*CAMK2D, OBSCN, LDB3*)[7]. In addition, this analysis identified a number of novel RBM20 targets including ion channels (*CACNA1C, KCNIP4, KCNQ5, SLC8A1*), RNA-binding proteins (RBPs) (*RBM20, FUS, QKI, FUBP3*), lncRNAs, and other cardiac regulatory factors (e.g., *MYOCD, CTNNB1, HAND2, MYH6, SPATS2, GSE1, GNAI3, TPM1*). Consistent with the literature, WT RBM20 bound principally to intronic elements and was most significantly associated with the established RBM20

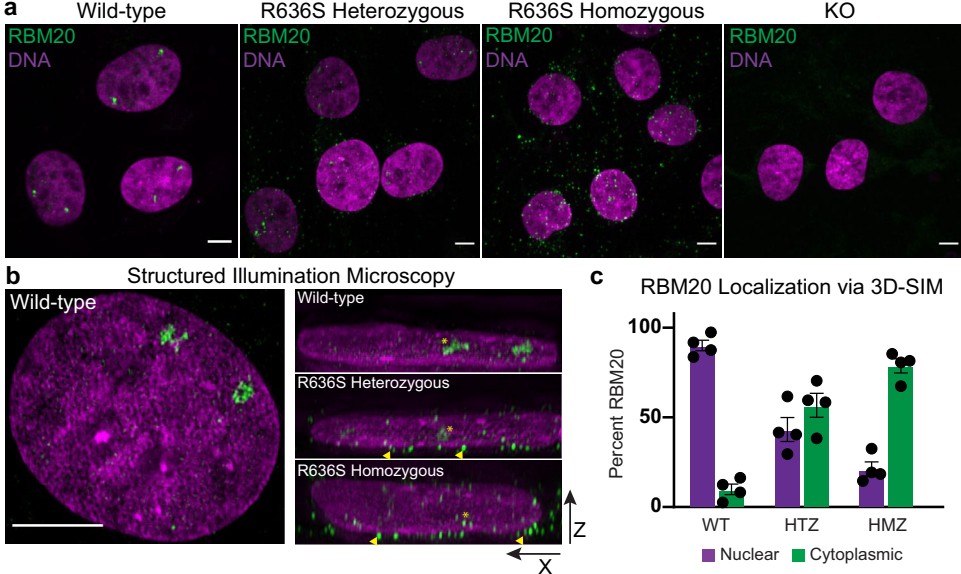

**Fig. 2 Intracellular RBM20 localization in RBM20 Mutant iPSC-CMs. a** Immunofluorescence for RBM20 in WT and RBM20 mutant iPSC-CMs. Spinning disk confocal, scale bar: 5 µm. Representative micrographs from *n* > 3 experiments. **b** Structured illumination microscopy of DNA (magenta) and RBM20 (green). Left WT image represents *X–Y* maximum intensity projection, right images represent *X–Z* 3D projections. Yellow arrowheads indicate perinuclear localization of RBM20 in mutant cells. Yellow stars indicate the nuclear localization of RBM20. Scale bar: 5 µm. Representative micrographs from *n* > 3 experiments. **c** 3D localization analysis of RBM20 from 3D-SIM images. Magenta bars (left bar above genotype) indicate percent nuclear localization, green bars (right bar above genotype) indicate percent cytoplasmic localization. Data represents per-cell averages from *N* = 2 independent experiments with *n* = 4 cells per genotype. Data represented as mean values ± SEM.

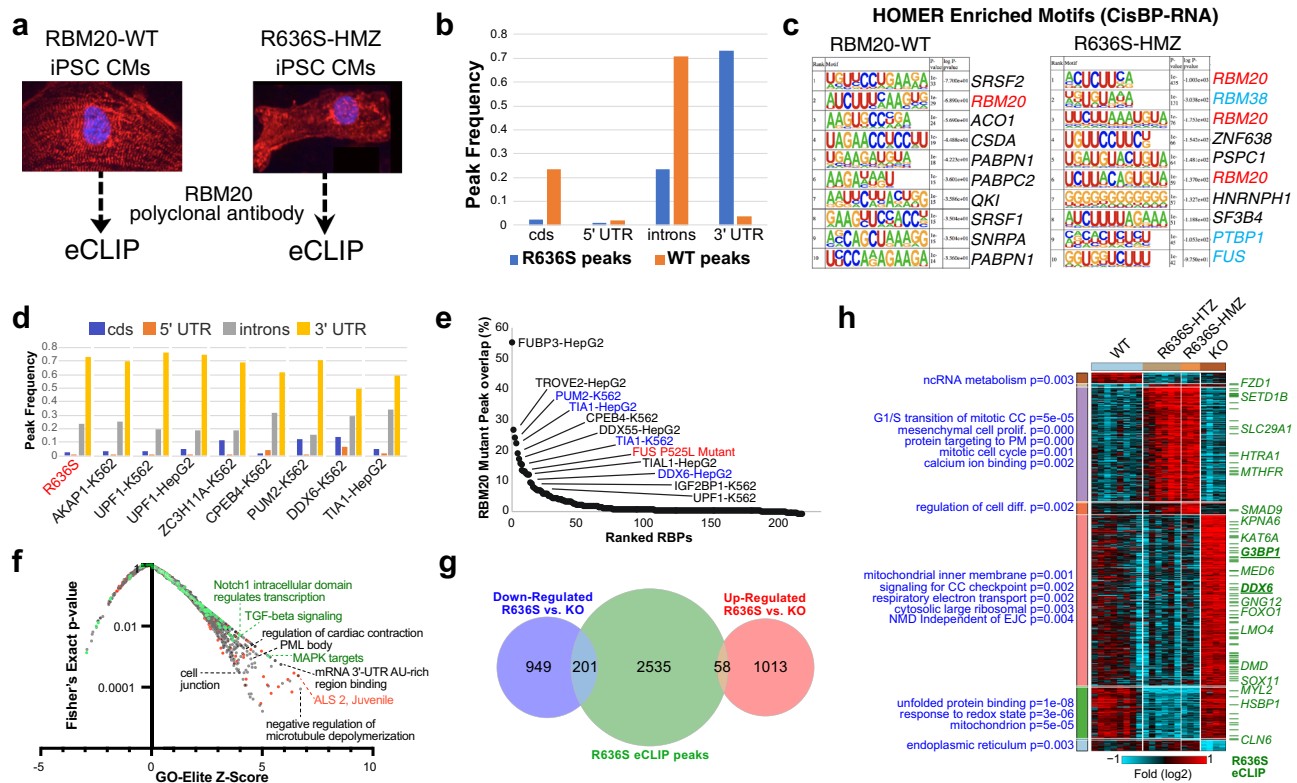

**Fig. 3 RBM20 mutant protein preferentially binds to the 3′ UTR of novel transcripts. a** Illustration of the eCLIP strategy for wild-type and R636S HMZ iPSC-CMs. **b** Frequency of reproducible eCLIP peaks for WT and R636S HMZ iPSC-CMs within coding sequence (cds) exons, introns and UTR regions. **c** HOMER de novo motif enrichment logos and hypergeometric (default) enrichment p-values for the top-ranked RNA-binding protein recognition elements defined from the CisBP-RNA database. RBM20-associated motifs are highlighted in red and RNA-stabilization-associated factors in blue. **d** Frequency bar chart of reproducible peaks in R636S HMZ eCLIP and the most similar eCLIP profiles from ENCODE (K562 or HepG2 cells), based on correlation to their frequency profiles. **e** Overlap of R636S peaks in reproducible ENCODE eCLIP peaks, ranked by their percentage overlap. One previously described ALS eCLIP profile for the FUS-P525L mutation is included and highlighted in red, representing peaks only found in the FUS mutant compared to controls. Blue text indicates RBPs with statistically enriched RBM20 motifs based on HOMER (Supplementary Fig. 4b). **f** Gene-set enrichment with GO-Elite (Fisher Exact test p < 0.05, raw) of genes associated with FUS-P525L and R636S overlapping peaks, for disease-associated gene-sets (red, DisGeNET), aggregate pathways (green, ToppFun), and Gene Ontology (black). **g** Overlap of genes up- or down-regulated by RNA-Seq in R636S iPSC-CMs (HTZ+HMZ) versus RBM20 deletion and R636S eCLIP peaks (eBayes two-sided t-test p < 0.05, FDR corrected). **h** Heatmap of all statistically ranked and organized (MarkerFinder algorithm) RBM20 R636S or deletion genes in iPSC-CMs by RNA-Seq. Statistically enriched gene-sets (GeneOntology + PathwayCommons) are indicated in blue with their associated Fisher's Exact test p-value (unadjusted) and R636S eCLIP peaks indicated by a green dash (selected genes shown). Source data are provided as a Source Data file.

consensus motif (UCUU) (Fig. 3b, c). In contrast, eCLIP of R636S HMZ iPSC-CMs identified 18,310 peaks in 2794 genes. Most notably, more than 70% of RBM20 bound sequences within R636S HMZ iPSC-CMs were within the 3′ UTR of transcripts (Fig. 3b, Supplementary Data 3) and overlapped WT peaks in only 29 genes (e.g., *CACNA1C, TTN, GNAI3, QKI*). While the canonical RBM20-binding site remained the most enriched RNA recognition element (RRE) for R636S, secondary enrichments were found for *PTBP1* and *FUS* binding sites, RBPs that are known to bind to the 3′ UTRs of transcripts in stress granules (Fig. 3c).

Given its cytoplasmic localization and previous indications that the R636S mutant interacts with stress granules[24], we hypothesized that RBM20 R6363S might act as an RNA stabilization factor. To test this hypothesis, we compared the transcript localization distribution of RBM20 R636S to all other RBP eCLIP profiles from ENCODE[42]. Indeed, the R636S eCLIP was most similar to the binding pattern of well-established processing-body (P-body)/stress-granule associated RBPs including *PUM2, DDX6,* and *TIA1*, all of which have eCLIP peaks enriched for the RBM20-binding site (Fig. 3d, Supplementary Fig. 2b). Direct comparison of R636S eCLIP peaks with all ENCODE eCLIP

profiles resulted in a similar distribution of overlapping RBPs, including *IGF2BP1* and *TIAL1* (Fig. 3e). Interestingly, mutant RBM20 eCLIP binding sites also overlapped with those of a mutant form of FUS associated with ALS (P525L) (Fig. 3e)[43]. While mutant RBM20 eCLIP peaks were enriched in RNA-stabilization and cardiac contractile genes (Supplementary Fig. 2c, d), common peaks from both FUS-P525L and RBM20-R636S were also enriched in 3′ UTR AU-rich binding (e.g., *ELAV1, ME3D, CPE3*), ALS (e.g., *SETX, DCTN3, TRAK2*), PML body, cardiac contraction, and cell junction genes (Fig. 3f, Supplementary Data 4, 5). To assess the potential functional significance of these RBM20-R636S-binding events, we performed deep RNA-Seq on WT (n = 8), R636S HTZ (n = 6), R636S HMZ (n = 3) and KO (n = 4) day 26 iPSC-CMs purified by sodium lactate treatment. Notably, R636S eCLIP peaks were relatively frequent (17%) among genes that were down-regulated in R636S versus RBM20 KO iPSC-CMs (Fig. 3g and Supplementary Data 6). Looking more broadly at global patterns of gene regulation identified striking differences between WT and KO or R636S iPSC-CMs, but only a few genes were consistently regulated in KO and R636S versus WT or between R636S HTZ and HMZ iPSC-CMs (Fig. 3h, Supplementary Fig. 2e and Supplementary

Data 7). Among genes with the dominant pattern of up-regulation in the KO and coincident down-regulation in R636S iPSC-CMs, genes with R636S eCLIP peaks were abundant, including those for RBPs (DDX6 and the IGF2BP1-binding partner G3BP1) (Fig. 3h). Taken together, these data indicate that R636S RBM20 shares common binding sites with stress granule/P-body RBPs and with ALS mutant FUS, many of which are down-regulated only with R636S.

**Alternative splicing differences in R636S and RBM20 KO CMs impact distinct physiological pathways.** Prior RBM20 KO studies in mice and rats, and patient RNA-Seq studies have identified a large number of RBM20-dependent splicing events implicated in DCM pathophysiology. To investigate the role of RBM20-mediated splicing in cardiac development, we first performed deep RNA-Seq of WT iPSCs at various stages of differentiation (Supplementary Fig. 3, S4 and Supplementary Data 8–13). To detect changes in global splice patterns, we used a combination of analysis software (AltAnalyze and MultiPath-PSI; see the "Methods" section for details), allowing for an unbiased and quantitative comparison of known and novel splicing events between time points[19,44,45]. In normal iPSC-CM differentiation, we observed an RBM20 splicing program induced in lock-step with an increase in *RBM20* gene expression (Supplementary Fig. 4j). These include well-established RBM20 target exons in *TTN* and *CAMK2D*. Notably, all novel cardiac differentiation splicing events tested ($n = 17$) were readily verified by RT-PCR or targeted sequencing, indicating that our splicing analysis predictions are highly accurate (Supplementary Fig. 4h, i and Supplementary Data 12). Having confirmed the sensitivity of these algorithms and the emergence of an endogenous RBM20 splicing program during differentiation, we performed a global splicing analysis in gene-edited RBM20 iPSC-CMs. In R636S HTZ, we identified 116 splicing events that differed from those in WT iPSC-CMs using a conservative LIMMA-based analysis (Fig. 4a; Supplementary Data 14; see the "Methods" section). These include several prior validated targets affected in DCM patients with the RBM20 S635A mutation, such as *TTN*, *CAMK2D*, *OBSCN*, *RYR2*, *IMMT*, and *TNNT2* (Fig. 4b–d, and Supplementary Fig. 5b)[16]. Several of these splicing events showed a clear dosage-dependent splicing pattern and overlap with RBM20 WT eCLIP peaks, suggesting that they are direct targets (e.g., *TTN*, *CDC14B*, *GSE1*, *RYR2*, *SPATS2*), while others had the same extent of splicing-deregulation with one or two alleles of R636S (e.g., *NEO1*, *IMMT*, *DMD*) (Fig. 4b, d, e). Approximately 80% of the RBM20 R636 HTZ alternative splicing events found were associated with cassette-exon splicing, in particular exon inclusion, consistent with prior reports (Fig. 4a)[7]. Notably, we found that >50% of these R636S-dependent splicing events were secondarily observed in independent RBM20 DCM mutant edited iPSC lines from our laboratory and others (Supplementary Fig. 5a; Supplementary Data 15–18). To assess the potential developmental regulation of these splicing events, we examined them in our WT iPSC-CM differentiation time-course. This analysis finds that R636S results in a developmental reversion of splicing for several well-described and novel RBM20 target exons (e.g., *CAMK2D*, *TTN*, *GSE1*) (Supplementary Fig. 4e, f; Supplementary Fig. 5a), along with others verified by RT-PCR (Supplementary Fig. 5i).

To identify predominant patterns of splicing in R636S mutation and RBM20 KO cells, we applied the same supervised-pattern analysis we applied to the gene expression data (Fig. 4c; Supplementary Data 19; see the "Methods" section). This analysis revealed stark differences in the patterns of splicing compared to gene expression. Specifically, we observed splicing

differences largely specific to the HTZ or HMZ R636S mutants, in addition to splicing differences uniquely shared by R636S HTZ and HMZ, those specific to RBM20 KO or shared among all R636S and RBM20 KO samples. The R636S/KO shared events frequently overlapped with WT eCLIP peaks and included the majority of prior validated RBM20 targets from the human heart and rat RBM20 truncation mutant studies[7,16]. As these splicing events co-occurred in KO cells, they likely are the result of the nuclear depletion of RBM20. Intriguingly, splicing-events specific to the R636S mutants or KO represented the dominant pattern of splicing and suggest that R636S does not simply phenocopy loss of RBM20 expression. Such R636S and KO-specific events included splicing regulators themselves (e.g., *LUC7L3*) as well as putative heart contractile disease mediators (*SLC44A2*, *SLCA81*, *SVIL*) (Fig. 4f–i; Supplementary Fig. 5). To determine whether such spliced exons are also present in an in vivo model of RBM20 DCM, we re-analyzed recent RNA-Seq from R636S gene-edited neonatal pigs, with HTZ and HMZ alleles[24]. This analysis verified splicing events within all observed splicing patterns that were conserved to a pig at the level of exon–exon junctions (Fig. 4c; Supplementary Fig. 6; Supplementary Data 20). R636S-unique splicing patterns were further evidenced by R636S intronic eCLIP binding sites as well independent iPSC-CM RNA-Seq, with RBM20 missense and non-sense mutations (Fig. 4c). Gene-set enrichment analysis of RBM20 splicing events that fell into the three major patterns (commonly regulated, specific to KO, specific to R636S), highlighted distinct cardiac contractility, physiology and signaling pathways that represent potential areas for targeted exploration in the future, including ion channel-specific splicing differences found only in R636S (e.g., *TRPC3*, *CACNA1G*, *CACNA1D*, *CACNB3*, *KCNH2*, *KCNC4*) (Fig. 4j and Supplementary Data 21). In summary, we find that R636S and RBM20 KO result in largely distinct changes in splicing and that such human iPSC-CM events are conserved in an R636S pig model of DCM.

**Mutant and KO RBM20 alter circular RNA production and alternative polyadenylation.** In addition to alternative splicing, recent studies suggest that circular RNAs, in particular those in *TTN*, may also contribute to RBM20-DCM pathology[46,47]. Using an RNA-Seq protocol sensitive to circRNAs we found that R636S mutants produce a far greater number of circRNAs than RBM20 KO iPSC-CMs (Supplementary Data 22). Consistent with prior studies, the most frequently detected circRNAs were found in *TTN*, with relatively few differential circRNA exons/introns containing RBM20 eCLIP binding sites or overlapping with alternatively spliced-exons (Supplementary Fig. 7a, b).

While not previously implicated in RBM20 biology, we further asked whether alternative polyadenylation (APA) occurred in the setting of mutant or KO KBM20 from our RNA-Seq. Analysis of differential APA events using the QAPA algorithm highlighted ~1200 events that were significantly associated with the dominant patterns of RBM20 mutant and KO regulation (Supplementary Fig. 7c and Supplementary Data 23). KO-induced APA events were enriched in mutant RBM20 eCLIP targets, which were predicted to mediate heart morphogenesis, the Golgi-network, cell-cycle, transcriptional regulation, cell-junctions, WNT, and BMP signaling, as well as 3′UTR AU-rich binding (Supplementary Fig. 7d, e and Supplementary Data 24). These data suggest that APA may represent a novel form of RBM20-dependent gene regulation that alters the ability of RBM20 to associate with 3′ UTR targets.

**R636S mutant RBM20 co-localizes with cytoplasmic processing bodies.** Based on our eCLIP comparative analyses, we predicted

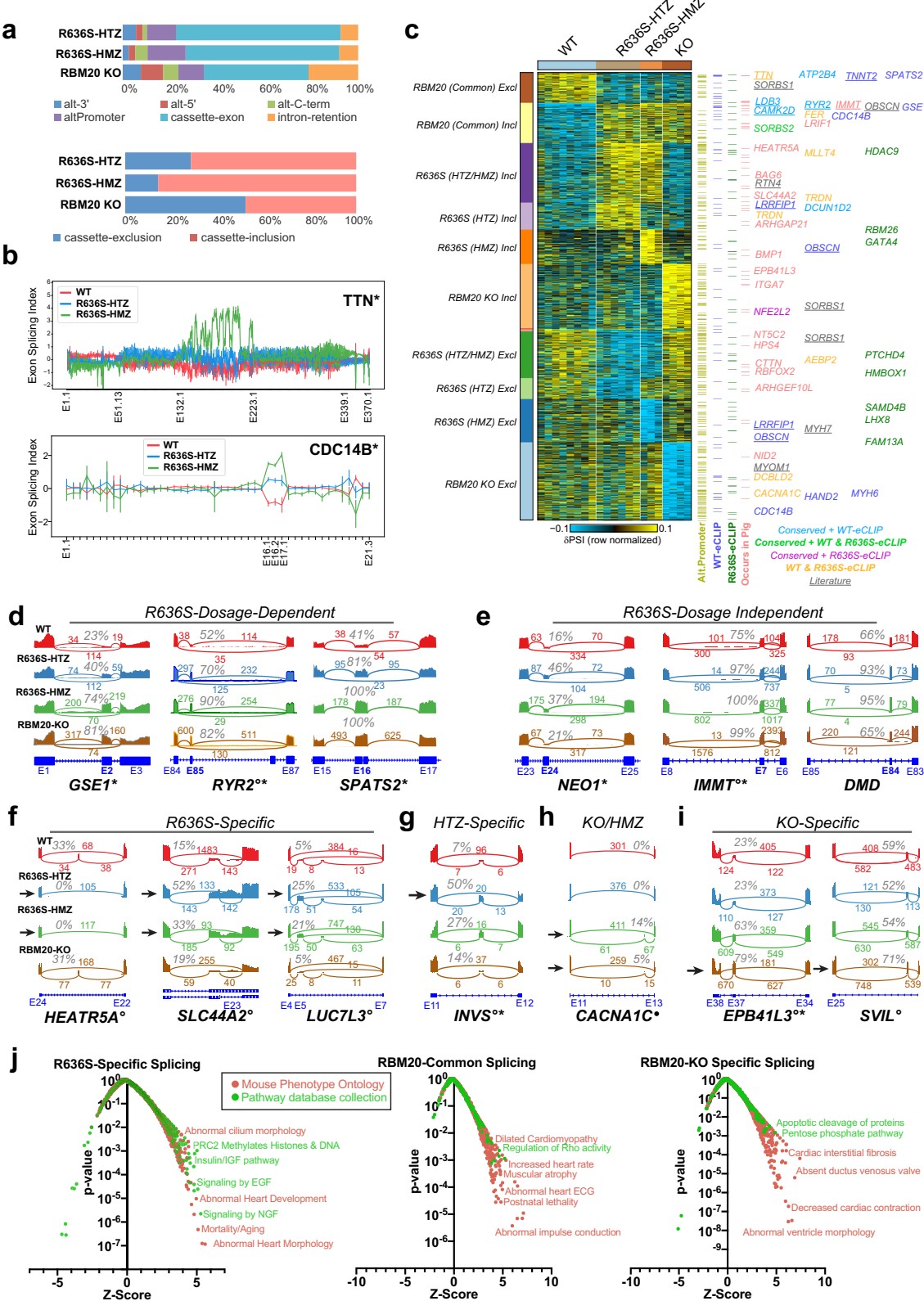

that mutant RBM20 may specifically localize within processing bodies (P-bodies), cytoplasmic ribonucleoprotein granules that exist in a liquid phase and may act to repress mRNA translation[48]. Importantly, P-bodies are distinct from stress granules, which are transient protein–RNA ensembles that assemble in response to acute stress, and disassemble when the stimulus is removed[49,50], although current models conflict on the

question of whether P-bodies and stress granules interact during acute stress[48,51,52]. The protein DDX6 was highlighted as one of the most likely RBM20 interacting P-body candidates from our analyses, as it shares predicted binding sites and binding profiles with RBM20 R636S. High-magnification imaging revealed that a subset of cytoplasmic RBM20 in mutant iPSC-CMs co-localized with P-bodies as visualized with DDX6 (Fig. 5a–c). RBM20 co-

**Fig. 4 RBM20 mutation and knockout impact distinct pathways at the level of alternative splicing. a** Percentage of alternative splicing events considered differential (LIMMA *t*-test *p* < 0.1, FDR corrected and δPSI > 0.1) for each of the iPSC-CM genotypes versus WT, separated by the predicted event-type (e.g., cassette-exon, intron retention) (top). Below, the percentage of splicing events associated with either cassette-exon inclusion or exclusion (skipping) are shown for each RBM20 genotype vs. WT. **b** Gene-level visualization of exon-level relative expression levels (splicing-index) for two of example significant genes, *TTN* and CDC14B, predicted to be alternatively spliced in R636S HTZ and HMZ cells in a dosage-dependent manner. AltAnalyze exon identifiers are shown below. **c** Heatmap of the predominant alternative splicing patterns (MarkerFinder) for all reasonably detected splicing events (eBayes two-sided *t*-test *p* < 0.05, δPSI > 0.1). The description of each pattern is displayed to the left of the heatmap (Incl exon-inclusion, Excl exon-exclusion associated splicing events). Splicing events with intronic eCLIP peaks in the same gene or that are also observed for the same exon–exon junctions as neonatal R636S porcine model (orthologous genome coordinates) are denoted to the right of the plot with examples listed. Underlined splicing-event genes indicate prior evidence of RBM20-dependent alternative splicing from rat KO studies. Associated statistics for all displayed events are provided in Supplementary Data 14–20. **d–i** SashimiPlot genome visualization of RBM20-dependent splicing events observed in iPSC-CMs, associating with distinct patterns of regulation. Representative samples were selected. Specifically, R636S-allele dosage-dependent splicing (**d**), dosage-independent R636S splicing (**e**), R636S but not KO-dependent splicing (**f**), R636S-HTZ-specific events (**g**), R636S-HMZ-specific events, and RBM20-KO specific events (**i**) among a series of those visualized by SashimiPlot analysis (see Supplementary Fig. 5 and S6). Splice-junction read counts are denoted above each curved exon–exon junction line, along with the estimated percentage of exon-inclusion. **j** Gene-set enrichment analysis (Fisher's Exact test *p*-value, unadjusted) of splicing-events segregated according to the MarkerFinder assigned patterns (panel **c**). Gene-sets correspond to either Mouse Phenotype Ontology or a collection of Pathway databases from ToppCell. *Verified splicing event patterns inferred from independently edited iPSC-CMs. °Verified splicing event from R636S HTZ edited pig hearts. ● R636S eCLIP intron bound peak containing. Source data are provided as a Source Data file.

localized with DDX6 to a greater degree in R636S HMZ than HTZ iPSC-CMs, and essentially no co-localization was observed in WT cells. Another P-body factor highlighted from our eCLIP analyses was IGF2BP1, which associates with the stress granule-promoting factor G3BP1. No stress granules were present in iPSC-CMs under basal culture conditions, in which G3BP1 displayed a diffuse cytoplasmic localization (Fig. 5d). However, following treatment with 1 mM sodium arsenate for 1 h, iPSC-CMs rapidly assembled conspicuous stress granules (Fig. 5d, e). Under these acute stress conditions, mutant RBM20 co-localized with stress granules, but WT protein did not (Fig. 5e). Hence, these data strongly support a model by which mutant RBM20 impacts alternative splicing through broad non-nuclear 3′ UTR binding and association with P-bodies, a mechanism that is reminiscent of neurological diseases associated with RNA-binding protein cytoplasmic aggregation (Fig. 6).

## Discussion

Our understanding of the molecular and physiological effects of RBM20 mutations has remained elusive largely due to a lack of appropriate models that are not confounded by heart disease or patient genetics. Here, we applied precision targeting of an RBM20 patient genetic mutation in human iPSC-CMs as a model system to evaluate the impact of specific patient alleles and RBM20 KO. This controlled system allowed us to delineate the impact of RNA biogenesis on excitation–contraction coupling attributable to a single-variant substitution. We demonstrated distinct phenotypes via MEA between R636S mutant and RBM20 KO iPSC-CMs. Importantly, although previously reported (mouse, rat, and pig) HTZ animal models of RBM20 DCM did not exhibit contractile defects, our RBM20 R636S 3D-EHTs (HTZ and HMZ) demonstrate a significant contractile phenotype compared to WT. Thus, this work establishes human iPSC-CM 3D-EHTs as a powerful and physiologically relevant model to study RBM20 DCM and therapeutic development.

Immunofluorescence experiments confirmed that the RBM20 R636S mutant protein mis-localizes to the cytoplasm. However, super-resolution microscopy revealed that a subset of mutant RBM20, even in R636S HMZ iPSC-CMs, maintains some capability to localize to the nucleus. Intriguingly, immuno-fluorescence showed that mutant RBM20 in mutant, but not WT iPSC-CMs co-localizes with P-bodies, cytoplasmic RNP granules present under basal conditions, that are sites of mRNA storage and turnover. During the preparation of this manuscript, the Schneider lab reported that mutant RBM20 co-localizes with

stress granules in cells treated with sodium arsenate[24]. Stress granules are distinct from P-bodies and are assembled in response to acute stress which inhibits translation (e.g., sodium arsenate, sorbitol, UV). Our data demonstrate that under basal culture conditions, WT and RBM20 mutant iPSC-CMs do not contain stress granules, but only assemble canonical stress granules in response to acute stress. Under these acute stress conditions, RBM20 in the mutant, but not WT iPSC-CMs, does co-localize with stress granules. Schneider's group also reported the liquid-like behavior of mutant RBM20 (e.g., RBM20 potentially undergoes a "liquid–liquid phase separation"). As P-bodies have been demonstrated to represent a bona-fide liquid–liquid phase separated compartment[53,54], association with P-bodies may drive this liquid-like behavior of mutant RBM20. Further investigation is warranted to better understand the functional consequences of the liquid-like behavior of cytoplasmic RBM20 on cellular physiology.

To understand the molecular consequences of mutant RBM20 in the nucleus and cytoplasm, we performed unbiased eCLIP profiling of WT and R636S iPSC-CMs. The results reveal a change in spliceosome-mediated target interactions in the setting of mutant RBM20 due to nuclear depletion of RBM20 and an altered bias towards consensus binding sites within the 3′ UTR of distinct transcripts. These 3′ UTR targets provide insight into the function of mutant RBM20, as many are shared with other RNA stabilization factors involved in pathogenic granule formation, including mutant FUS[55]. These data provide strong evidence that mutant RBM20 mimics previously described factors that mediate P-body and stress granule formation, and it may recruit additional factors to these bodies.

Importantly, our alternative splicing predictions further suggest a more complex model of the role of mutant RBM20 in cardiomyopathy pathogenesis. Specifically, although R636S RBM20 molecules are largely absent from the nucleus, we observed highly contrasting splicing signatures between KO and HMZ mutant cells. The importance of these mutation-specific splicing events, the targets of which include genes involved in cardiac development, and structural and contractile regulators, are supported by the finding of orthologous R636S-induced events in neonatal pigs and/or that exhibit overlapping mutant-specific eCLIP peaks. Additional observations from circular RNA analyses indicate that mutant RBM20 affects circular RNA production, supporting prior observations that such molecular events preferentially impact constitutive exons, while alternative poly-adenylation analyses suggest a broader role RBM20 in post-

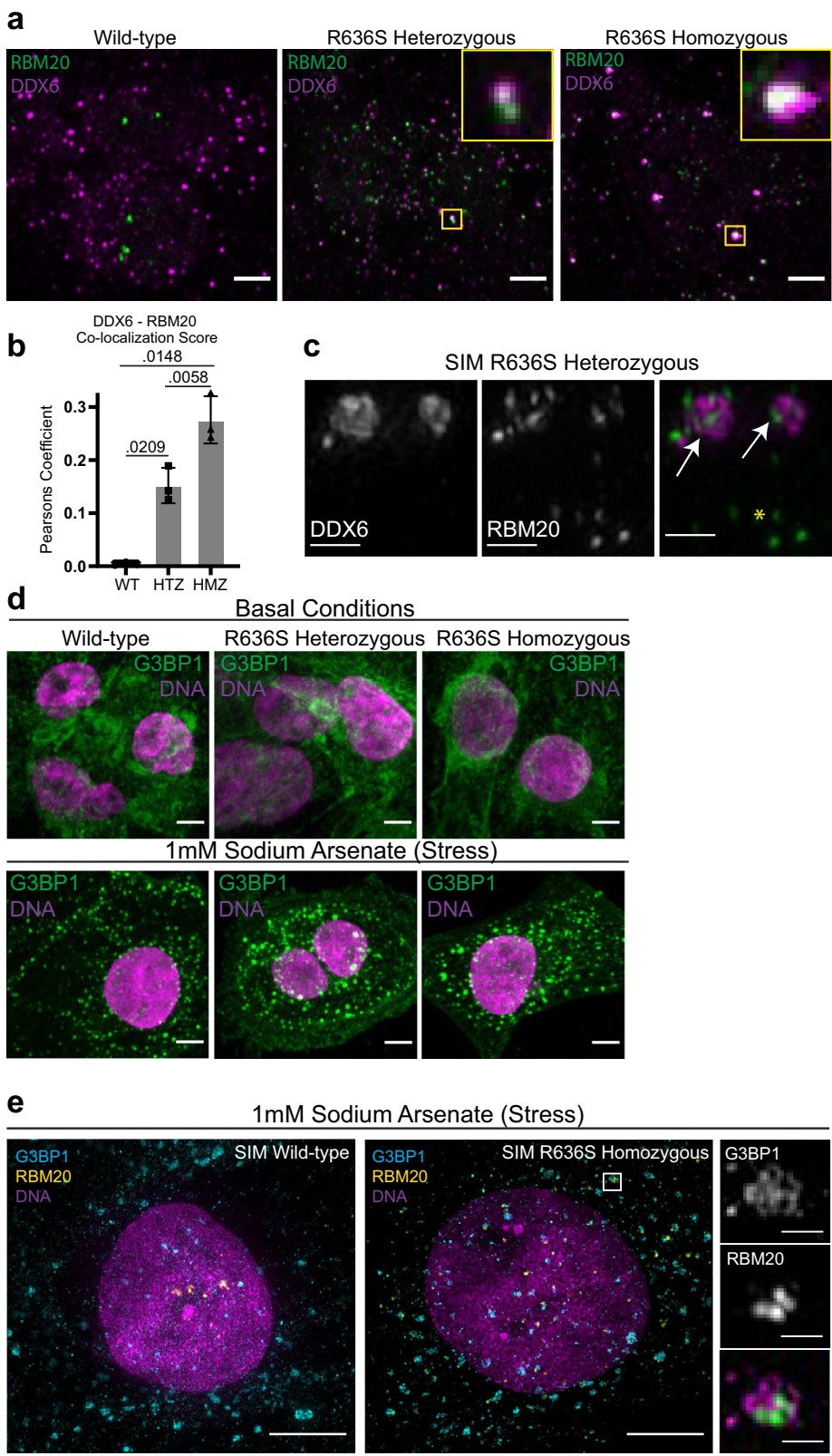

transcriptional gene regulation[46]. While the precise cause of and functional relevance of splicing differences in mutant and WT remain to be determined, these data suggest a more complex model in which loss of canonical RBM20 splicing leads to altered functional products involved in cardiac contractile regulation, potentially augmented by pathogenic circular RNAs, the emergence of new splice-forms in cardiac developmental and signaling genes, global changes in polyadenylation, and the degenerative accumulation of cytoplasmic granules impacting cardiomyocyte homeostasis.

It is intriguing to note the parallels between our observations with RBM20 and recent findings in neuro-degeneration. Indeed,

**Fig. 5 Intracellular localization of RBM20 mutant iPSC-CMs and association with P-bodies. a** Localization of RBM20 (green) and DDX6 (processing bodies, magenta). High-magnification insets highlight co-localization. Spinning disk confocal, scale bar: 5 μm. Representative micrographs from $n > 3$ independent biological experiments. **b** DDX6/RBM20 co-localization is quantified via Pearson's coefficient of co-localization. Data from $N = 3$ biological independent experiments. Statistical significance was calculated using one-way ANOVA with Dunnett's multiple comparisons test. Data represented as mean values ± SEM. **c** Structured Illumination microscopy of DDX6 (left), RBM20 (middle), and merged channels (DDX6: magenta, RBM20: green). White arrows indicate co-localization; yellow asterisk indicates RBM20 not associated with processing bodies. Scale bar: 0.5 μm. Representative micrographs from $n = 2$ independent biological experiments. **d** Assessment of stress granules via G3BP1 localization. iPSC-CMs were analyzed in basal conditions (e.g., normal media) and in response to stress (1 mM sodium Arsenate treatment for 1 h). Spinning disk confocal, scale bar: 5 μm. Representative micrographs from $n > 3$ independent biological experiments. **e** Structured illumination microscopy analysis of RBM20 (yellow) and G3BP1 stress granules (cyan) in sodium-arsenate-treated iPSC-CMs. Scale bar: 5 μm. Representative micrographs from $n = 2$ independent biological experiments. Source data are provided as a Source Data file.

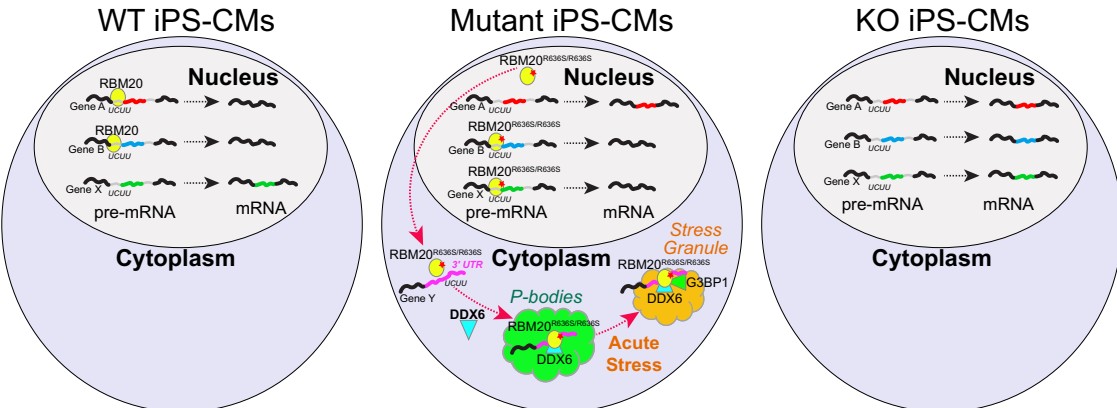

**Fig. 6 Model of mutant RBM20 differential splicing and P-body impacts in dilated cardiomyopathy.** Proposed model for the impact of wild-type and mutant RBM20 on nuclear regulation of splicing, based on RNA-Seq and eCLIP data, as compared to the cytoplasmic role of mutant RBM20 on P-body formation and 3′UTR association with mRNAs implicated in granule formation.

recent work has hypothesized cytoplasmic RBM20 may be similar to the cytoplasmic RNP granules associated with neurodegeneration[24], such as TAU for Alzheimer's disease, Huntingtin for Huntington' disease, and FUS for amyotrophic lateral sclerosis (ALS)[56,57]. We add to this initial hypothesis by demonstrating that under basal conditions, mutant RBM20 binds to 3′ UTR regions of mRNA and co-localizes with P-bodies. Our study indicates that protein granules are a critical determinant factor for not only neurodegenerative disease but potentially more diverse diseases including cardiomyopathy. This is an intriguing notion, but further mechanistic studies are required to determine the functional consequences of the cytoplasmic localization and association of mutant RBM20 with P-bodies.

These analyses illustrate important differences in the splicing targets of mutant RBM20, with data from iPSC-CMs likely indicating that defects during early cardiogenesis could lead to remodeling of the fetal heart. While such molecular changes in iPSC-CMs may not recapitulate the gross-cardiac remodeling defects that impact heart function in patients with RBM20 mutations, we hypothesize that such splicing differences may predispose to later cardiovascular events. Importantly, these human cellular models provide the cardiovascular research community with powerful tools to aid in the development of new therapeutic targets for heart disease.

## Methods

**iPSC culture**. The UCSF Committee on human research #10-02521 approved the study protocol for iPSCs. The human iPSC lines used in this study were generated from a healthy male patient, WTC11[26,58] using the episomal reprogramming method[59]. Informed consent was obtained for this procedure. iPSCs were maintained on Matrigel (BD Biosciences) in Essential 8 medium (Life Technologies), which was exchanged every other day. To sparsely populated wells (i.e., passaged wells), we added 10 μM Y-27632, a Rho-associated kinase (ROCK) inhibitor

(Millipore), to promote cell survival. WTB iPSCs were generated as described in ref. [60].

**RBM20 mutant iPSC line generation**. We applied our ddPCR and sib-selection-based strategy to isolate genome-edited iPSC lines, as previously described[25,26]. We used TALENs targeting exon 9 of RBM20 (Addgene #108342 and #108343) to generate the RBM20 R636S Het iPSC line, which was then retargeted by CRISPR/Cas9 nickase (pX335; Addgene #42335) to convert the WT allele to the R636S with an S635S silent mutation allele. We used the Human Stem Cell Nucleofector Kit-1 and a Nucleofector 2b Device (Lonza) to transfect the plasmids and oligonucleotide donor DNA. For each transfection, 2 million cells were transduced with 3 μg of each TALEN vector or pX335 and 6 μg of an oligonucleotide donor DNA using program A-23. The transfected cells were plated into a Matrigel-coated 96-well plate using a multichannel micropipetter. The detailed information of the TALENs, gRNAs, and oligonucleotide donor DNA are summarized in Supplemental Methods. Karyotyping was performed by Cell Line Genetics.

The composition of the premixtures of allele-specific TaqMan probes and primers for ddPCR analysis was 5 μM of an allele-specific FAM or VIC TaqMan MGB probe (Thermo Fisher Scientific), 18 μM of a forward primer, and 18 μM of a reverse primer (Integrated DNA Technology) in water. To detect point mutagenesis, we mixed the following reagents in 0.2 ml PCR 8-tube strips: 4 μl water, 12.5 μl 2× ddPCR Supermix for probes (Bio-Rad), 1.25 μl R636S + SM FAM probe and primer premixture, 0.625 μl WT VIC probe and primer premixture, 0.625 μl R636S FAM probe and primer premixture, and 5 μl (50–150 ng) genomic DNA solution (25 μl total volume). The conditions for droplet generation, thermal cycling, and data analysis for RBM20 R636S mutagenesis with the ddPCR system were described before[25,26]. As the R636S + SM FAM probe had a higher concentration than the WT FAM probe, the signal of the R636S + SM allele was distinguishable from that of the WT allele (Supplementary Fig. 1). Cell populations with a higher frequency of the R636S + SM allele and a lower frequency of the WT allele were enriched by sib-selection until the RBM20 R636S HMZ iPSC clone was isolated.

**iPSC-CM differentiation**. For genomic analyses (Figs. 1, 3, and 4) the iPSC lines were differentiated into iPSC-CMs using a method that controls WNT signaling by small molecules (the GiWi protocol)[61]. Briefly, iPSCs were seeded at $1.25–2.5 \times 10^4$ cells/cm² onto 12-well plates coated with 80 μg/μl growth factor-reduced Matrigel (BD Biosciences) in mTeSR1 supplemented with 10 μM Y-27632 (Millipore) for 24 h (day 3). mTeSR1 medium was changed daily for the next 2 days. On day 0,

iPSCs were treated with 12 μM CHIR99021 (CHIR) (Tocris) in RPMI supplemented with B-27 (RPMI/B27) without insulin (Life Technologies) for exactly 24 h. On day 1, the culture medium was replaced with fresh RPMI/B27 without insulin and maintained for 48 h. On day 3, cells were treated with 5 μM IWP2 (Tocris) in RPMI/B27 without insulin and maintained for 48 h. On day 5, fresh RPMI/B27 without insulin was added to the cells, and on day 7, the medium was switched to RPMI/B27 with insulin. Afterward, fresh RPMI/B27 with insulin was added to the cells every 3 days. We used a metabolic selection protocol with glucose-free DMEM containing lactate to purify iPSC-CMs. Cells were replated on day 15–18 of differentiation, and then on day, 20–22 media were replaced with DMEM (without glucose, with sodium pyruvate, Thermo Fisher Scientific) supplemented with Glutamax, non-essential amino acids, and buffered lactate (4 mM). Stock-buffered lactate solution was prepared by dissolving sodium L-lactate powder (Sigma-Aldrich) at 1 M concentration in 1 M HEPES solution. Lactate media were exchanged 2–3 times, with a total exposure of 48 h for each treatment. After the final lactate treatment, media were changed to RPMI/B27 with insulin. The purified WT or RBM20 mutant iPSC-CMs at day 26 were harvested for RNA-Seq or cryopreserved for further analyses.

For electrophysiological, contractile, and immunofluorescence analyses, iPSCs were maintained on Matrigel (Fisher Scientific) and in mTesr Plus (StemCell Technologies) culture media, which was exchanged every other day. For iPSC-CM differentiation, iPSCs were plated in 12-well plates (Fisher Scientific) coated with Matrigel in mTesr Plus supplemented with 10 μM Y-27632 (StemCell Technologies) for 24 h. On day −1, iPSCs were treated with 1 μM CHIR-99021 (Cayman Chemical) to prime cells for differentiation. On day 0, iPSCs were treated with RPMI-1640 with glutamine supplemented with 500 μg/ml BSA, 213 μg/ml ascorbic acid (RBA media; Millipore Sigma), and 5 μM CHIR-99021 for 48 h. On day 2, cells were treated with RBA media supplemented with 2 μM WNT-C59 (Selleck Chemicals). On day 4, cells were treated with RBA. Starting on day 6, iPSC-CMs were maintained in RPMI + B27 (ThermoFisher) with media exchanges every 2 days. On day 13, iPSC-CMs were pre-conditioned with a 45-min heat shock at 42 °C (pre-warmed 42 °C media was exchanged prior to incubation) to enhance the survival of cryopreservation and thawing process. On day 14, iPSC-CMs were frozen in batches (>5 million cells) in CryoStor media (1 million cells/100 μL CryoStor; Millipore Sigma) and placed in liquid nitrogen for disease modeling experiments. To passage cells for freezing, cells were washed with 1× DPBS (no magnesium, no calcium; ThermoFisher), and treated with 10× TrypLE (ThermoFisher) for 10–20 min at 37 °C. Cells were dissociated with mild trituration and passed through a 100 μM strainer (Fisher Scientific) to remove any large clumps (which negatively impact survival in our hands).

Frozen iPSC-CM stocks were thawed onto six-well plates coated with Matrigel at 2.5–3 million cells/well in RPMI- B27 supplemented with 10 μM Y-27632 and 10% FBS. 24 h post-thaw, media was exchanged for RPMI-B27. After 48 h recovery, iPSC-CMs were purified in no glucose DMEM + 4 mM sodium lactate to select for iPSC-CMs for a total of 4 days (media was exchanged on day 2 of treatment). After 4 days, media was exchanged with RPMI-B27, and iPSC-CMs were allowed to recover for one day before re-platting (as above) for subsequent studies at day 21 post-differentiation.

**Flow-cytometry**. Day 15 cells were dissociated into single cells using 0.2 ml of 0.25% Trypsin (UCSF Cell Culture Facility) per well of a 24-well plate with incubation at 37 °C. Cells were resuspended in 0.8 ml/well EB medium (KO DMEM supplemented with 20% FBS (HyClone), Glutamax (Gibco), non-essential amino acids (UCSF Cell Culture Facility), and 0.1 mM beta-Mercaptoethanol (Sigma)). 200 μL (about 200,000) cells were pelleted and fixed in 4% paraformaldehyde at room temperature for 15 min. Cells were permeabilized in FACS buffer (DPBS without calcium and magnesium, 4% FBS, and 2 mM EDTA (Gibco)) with 0.5% (w/v) saponin (Sigma). Cells were stained with 100 μl of 0.002 μg/μl (1:100) Mouse anti-human cardiac Troponin T (cTnT) primary antibody (Thermo, MS-295-P) in FACS buffer with saponin at room temperature for 30 min and washed. Cells were then stained with 100 μl of 0.01 μg/μl (1:200) Alexa Fluor 488 goat anti-mouse IgG secondary antibody (Invitrogen, A-11029) in FACS Buffer with saponin at room temperature for 30 min and washed. Finally, cells were stained with 100 μl of 0.01 μg/μl (1:1000) Hoechst 33342 (Molecular Probes) in FACS buffer at room temperature for 5 min and passed through a 0.4 μm filter (Millipore) to remove cell clumps. Data were collected using the MACSQuant VYB flow cytometer (Miltenyi Biotec) and analyzed using FlowJo.

**RNA-sequencing**. Total RNA was purified from iPSC-CMs by using TRIzol and PureLink RNA Mini Kit (Thermo Fisher Scientific) according to the manufacturer's instructions. The RNA-Seq libraries were prepared from the purified RNA by using TruSeq RNA Library Prep Kit v2 (Illumina) for gene and splicing analyses or the RiboMinus for Ribosomal RNA Depletion protocol (Thermofisher) for circular RNA splicing analyses. Paired-end RNA-Seq was performed with HiSeq (Illumina). An average of ~90 million reads was obtained from this iPSC RNA-Seq data. RNA-Seq for WTB iPS differentiation was performed as biological replicates (n = 2), whereas the number of samples processed for WTC iPSC-CMs was 3 to 7 replicate samples per genotype.

**eCLIP**. RBM20 WT and RM20-R636S direct transcript-binding sites were determined using the same RBM20 polyclonal antibody (ThermoFisher, Cat#PA5-58068) on WT or R636S HMZ WTC iPS CMs using the eCLIP protocol as previously described[41]. RNA-libraries were sequenced at a depth of 30–40 million reads for each replicate eCLIP sample, on an Illumina HiSeq 2500 (single-end reads). Reproducible RBM20 peaks (hg19) obtained from replicate WT and R636S HMZ iPSC-CMs compared to size-matched input controls, were used for all downstream analyses. eCLIP peaks (hg19) from all prior generated ENCODE RBPs (K562 and HEPG2 cell lines) were obtained from https://www.encodeproject.org/eclip/. Additionally, peaks associated with the ALS mutation in the gene FUS (P525L) were aggregated within a 200nt window (merge function), only found in the mutant and not the FLAG-tagged eCLIP, following LiftOver (http://genome.ucsc.edu/cgi-bin/hgLiftOver) from hg38 to hg19 (GSE118347). To identify RREs associated with known RBPs, we analyzed reproducible RBM20 peaks using the software HOMER using de novo motif discovery with the CisBP-RNA database[62,63]. All eCLIP peaks were annotated according to the gene intervals they correspond to (e.g., exon, introns, 3′ ends, 3′ UTR, distal or proximal regions) using predictions produced from clipper (https://github.com/YeoLab/clipper/). eCLIP peaks overlapping between different RBPs were identified using the bedtools intersect -wb option.

**RNA-Seq analysis**. RNA-Seq FASTQ files were aligned to the human hg19 reference genome and transcriptome using the software STAR. STAR-produced BAM files were further processed in AltAnalyze (version 2.1.4) to obtain gene expression (RPKM values) and splicing estimates (PSI). Gene expression differences across the 11 CM differentiation time-points were calculated as pairwise comparisons to day 0 and to the prior time-point, for gene-set enrichment analysis and to restrict genes for pattern-based analyses (MarkerFinder algorithm in AltAnalyze) using an emperical Bayes two-sided t-test $p < 0.05$ (FDR corrected) and at least a 2-fold difference with a minimum RPKM value of 1 in either of the two groups compared. To accurately detect diverse alternative splicing events in iPSC-CM RNA-Seq, we used a recently described Percent Spliced-In splicing approach, called MultiPath-PSI, capable of accurate detection of known and novel exonic and intronic splicing events, as well as alternative promoter-associated exons[8,45,64]. Alternative polyadenylation was predicted and quantified using the software QAPA with default options to produce alternative isoform ratios for each 3′ UTR isoform, normalized to the total 3′ UTR form for each gene[65].

Application of MultiPath-PSI to all evaluated mutant and matched human WT samples identified 44,911 detected splicing events, corresponding to 16,207 unique splicing events. Prior to differential splicing analysis, empirically observed gene expression associated differentiation effects were evaluated using an unsupervised subtype detection analysis with the software Iterative Clustering and Guide-gene Selection (ICGS)[66]. Since the identified transcriptomic differences among independent iPSC-CM differentiations could not be associated with iPSC genetic background, we surmised these are likely due to differences in cardiac maturity. To account for these empirically observed effects, we applied a mixed-effects linear model. For this model, each pair of differentiation-associated transcriptional effects was evaluated with LIMMA using the lmfit function to account for these effects in addition to cell line genetics in the linear model.

For differential splicing analyses, splicing events were excluded that were not detected in at least 75% of the samples (i.e., gene expression not detected) in each biological group along with a change in PSI between the two groups >10% (δPSI > 0.1). The supervised splicing pattern analysis was performed on unique differential splicing events (splicing-event clusterID) using the R package LIMMA to obtain a two-sided emperical Bayes t-test p-value for the following models: (1) R636S HTZ-specific splicing, (2) R636S HMZ-specific splicing, (3) equivalent differential splicing in R636S HTZ and HMZ, and (4) additive induction in the HTZ to HMZ R636S. To identify splicing events associated with the predominant genetically defined patterns, we applied the MarkerFinder algorithm, iterated over multiple sample groupings (all primary groups and aggregated groups of mutations —e.g., R636S-HTZ plus R636S-HMZ), for empirical Bayes two-sided t-test significant events (p < 0.05). This algorithm was also used for gene expression and alternative polyadenylation analyses, to define the major RBM20-dependent patterns (see iterativeMarkerFinder function from https://git.io/Jt9je). Alternative splicing events from independently edited WTB iPSC-CMs with R636S HTZ and controls were quantified as above by TruSeq RNA-Seq, followed by AltAnalyze and MultiPath-PSI (δPSI > 0.1). Similarly, RNA-Seq FASTQ files from a prior study were downloaded from PRJNA579336, to identify differential splicing results associated with missense or nonsense HMZ RBM20 mutations and controls, using AltAnalyze and MultiPath-PSI (δPSI > 0.1 and empirical Bayes two-sided t-test $p < 0.05$)[67]. Gene-set enrichment analyses were performed with the software GO-Elite in AltAnalyze[68].

To identify, quantify and annotate circular RNAs from back-splice junctions we first aligned the RNA-Seq results using TopHat2 on the RiboMinus depleted paired-end RNA-Seq FASTQ files, via the CIRCexplorer2 pipeline, on each individual sample (default options). This workflow identified far greater putative circRNAs than using the same pipeline with STAR 2.4.0. For differential circRNA analysis we ran RUVSEQ from the CSBB pipeline (https://github.com/csbbcompbio/CSBB-v3.0) on the back-splice junction counts for each sample for all relevant comparison groups. This workflow applies the software EdgeR to perform

a differential expression analysis using a generalized linear model approach using the upper quartile normalization. All together, we identified circRNAs for 448 genes, with evidence of at least two reads per circRNA. circRNAs with a non-adjusted EdgeR $p < 0.1$ were considered differentially expressed.

To determine genomic coordinate overlaps between: (a) eCLIP peaks and alternative splicing events, (b) eCLIP peaks and alternatively expressed circRNAs, or (c) alternative splicing events and alternatively expressed circRNAs, we created a multi-use script that leverages transcriptomic feature genomics coordinates and their overlap in AltAnalyze (https://git.io/JtHfa). For alternative splicing, the MultiPath-PSI reported alternative exon and flanking intron genomic positions were directly intersected against eCLIP peaks. Similarly, the regulated exon(s) and flanking introns were considered for overlap for circRNAs.

**Quantitative PCR analysis**. Quantitative PCR gene expression analyses was performed on the total RNA isolated during WTB iPS cell differentiation into CMs. cDNA was generated from 1 μg of TurboDNAse-treated (Ambion) total RNA with the SuperScript III First Strand Synthesis kit and random hexamers (Invitrogen) as described by the manufacturer. Expression was assessed using TaqMan probesets run on the 7900HT real-time thermocycler (Applied Biosystems). Samples were assayed in technical triplicate, normalized to GAPDH or UBC, and relative expression was calculated with the day of highest expression set to 100%.

**Automated primer design and RT-PCR analysis of ASEs**. To validate alternatively spliced exons, the primer design program Primer3[69] was integrated with AltAnalyze (identifyPCRregions function in the EnsemblImport module). As sequence input, the alternative spliced exon, junctions, and associated isoforms were used to identify the target region in the inclusion isoform (included/excluded exon), and suitable upstream and downstream exon sequences shared by the two isoforms. RT-PCR was conducted using these primers (Supplementary Data 12), random hexamer generated cDNA (described above), and Platinum Taq DNA Polymerase High Fidelity (Life Technologies) as described by the manufacturer. For an example of the presentation of full scan blots, see the corresponding Source Data files.

**Engineered heart tissue casting**. 3D-EHTs were cast and characterized as previously described[31]. Polydimethylsiloxane (PDMS) posts were fabricated by pouring uncured PDMS (Sylgard 184 mixed at a 1:10 curing agent to base ratio) into a custom acrylic mold (Limited Productions Inc., Bellevue, WA; design available upon request). Glass capillary tubes (1.1 mm in diameter; Drummond) were cut to length and inserted into the holes on one side of the mold before curing to render one post in each pair rigid. Post racks were baked overnight at 65 °C before being peeled from the molds. Racks consisted of six pairs of posts that were evenly spaced to fit along one row of a standard 24-well plate. Fabricated posts were 12.5 mm long and 1.5 mm in diameter with a cap structure (2.0 mm in diameter for the topmost 0.5 mm) to aid in the attachment of 3D-EHTs. The center-to-center post spacing (corresponding to pre-compacted 3D-EHT length) was 8 mm. Prior to casting 3D-EHTs, all 3D-printed parts and PDMS posts were sterilized in a UVO Cleaner (342; Jetlight) for 7 min, submerged in 70% ethanol, and rinsed with sterile deionized water. Rectangular 2% wt/vol agarose/PBS casting troughs (12 mm in length, 4 mm in width, and ~4 mm in depth) were generated in the bottom of 24-well plates by using custom 3D-printed spacers (12 mm × 4 mm in cross-section and 13 mm long) as negative molds. PDMS post racks were positioned upside down with one rigid-flexible post pair centered in each trough (leaving a 0.5-mm gap between the tip of the post and the bottom of the casting trough). Each tissue consisted of a 97-μl fibrinogen-media solution (89 μl of RPMI-B27, 5.5 μl of DMEM/F12, and 2.5 μl of 200 mg/ml bovine fibrinogen; Sigma-Aldrich) containing $5 \times 10^5$ iPSC-CMs and $5 \times 10^4$ supporting HS27a human bone marrow stromal cells (ATCC), which was chilled and mixed with 3 μl of cold thrombin (at 100 U/ml; Sigma-Aldrich) just before pipetting into the agarose casting troughs. The 3D-EHT mixtures were incubated for 90 min at 37 °C, at which point the fibrin gels were sufficiently polymerized around the posts to be lubricated in media and transferred from the casting troughs into a 24-well plate with fresh 3D-EHT media (RPMI-B27 with penicillin/streptomycin, and 5 mg/ml aminocaproic acid; Sigma-Aldrich). 3D-EHTs were supplied with 2.5 ml/well of fresh 3D-EHT media three times per week.

In situ contractile measurements were performed at 3 weeks post-casting of 3D-EHTs. To pace 3D-EHTs, post racks were transferred to a custom-built 24-well plate with carbon electrodes connected through an electrical stimulator (S88X; Astro Med Grass Stimulator) to provide biphasic field stimulation (5 V/cm for 20-ms durations) during imaging (Leonard et al., 2018). 3D-EHTs were equilibrated in Tyrode's buffer (containing 1.8 mM $Ca^{2+}$) preheated to 37 °C and paced at 2 Hz, which was greater than the average spontaneous twitch frequency of the tissues. Videos of at least 10 contractions were recorded inside a 37 °C heated chamber using a monochrome CMOS camera (ORCA-Flash4.0). The camera lens configuration allowed for a capture rate of 66 fps with 8.3 μm/pixel resolution and a FOV of 1536 × 400 pixels, which was sufficient to capture images of the whole 3D-EHT from rigid to flexible post. A custom Matlab program was used to threshold the images and track the centroid of the flexible post relative to the centroid of the rigid post. The twitch force profile, $F_{twitch}(t) = k_{post} \times \Delta_{post}(t)$, was calculated from

the bending stiffness $k_{post}$ and deflection of the flexible post $\Delta_{post}$ at all time points ($t$), where $k_{post} = 0.95$ μN/μm was determined from beam bending theory using the dimensions of the posts and taking Young's modulus of PDMS to be 2.5 MPa[70]. The twitch force and twitch kinetics were calculated from the twitch force profiles using a custom Matlab program. For statistical significance testing, one-way ANOVA with a Dunnett's correction for multiple comparisons was performed.

**Multi-electrode array (MEA)**. iPSC-CMs were plated (as above) at 100,000 cells per well directly on the electrode array of 24-well MEA plates (CytoView MEA 24, Axion Biosystems) pre-coated with matrigel. RPMI-B27 was exchanged every 2 days. MEA analysis occurred at day 35 post differentiation (2 weeks post-plating on MEA plates). MEA data was acquired at 37 °C and 5% $CO_2$ and recordings were acquired for 5 min using the Maestro MEA system (Axion Biosystems) using standard recording settings for spontaneous cardiac field potentials. Automated data analysis was focused on the 30 most stable beats within the recording period. The beat detection threshold was dependent on the individual experiment, and the FPD was manually annotated to detect the T-wave. The FPD was corrected for the beat period according to Fridericia's formula: FPDc = FPD/(beat period)$^{1/3}$ [71–73]. Results for individual wells were calculated by averaging all of the electrodes. For each MEA experiment, 4 technical replicates per condition were averaged and reported results represent the average of three distinct biological replicates. For statistical significance testing, one-way ANOVA with a Dunnett's correction for multiple comparisons was performed.

**Fixation and immunohistochemistry**. For images in Figs. 2 and 5, iPSC-CMs were fixed with 4% paraformaldehyde (PFA) in PBS at room temperature for 20 min and then extracted for 5 min with 0.3% Triton X-100 and 4% PFA in PBS as previously described (Burnette et al., 2014). Cells were washed three times in 1 × PBS. Cells were blocked in 5% BSA in PBS for at least 30 min. Primary antibodies were diluted in 5% BSA + 0.3% Triton X-100. RBM20 antibody (ThermoFisher, Cat#PA5-58068) was used at 1:500, DDX6 (MilliporeSigma, Cat# SAB4200837), and G3BP1 (SantaCruz Biotechnology, Cat# sc-365338) antibodies were used at 1:200. For actin visualization, phalloidin-488 (ThermoFisher Scientific, Cat# A12379) in 1x PBS (15 μl of stock phalloidin per 200 μl of PBS) was used for 3 h at room temperature. DNA was visualized via DAPI staining for 30 min at room temperature (final concentration 1.2 μM). Cells were kept and imaged in 1× PBS.

**Structured illumination microscopy**. SIM microscopy was performed on a GE Healthcare DeltaVision OMX SR microscope equipped with a PLAPON ×60OPSF/1.42 NA objective with a pco.edge 4.2 sCMOS camera at room temperature. Image reconstruction was performed using GE softWoRX software. Grant S10 OD021490

**Spinning disk microscopy**. Spinning disk microscopy was performed on a Nikon Eclipse Ti equipped with a Yokogawa CSU-W1 spinning disk head, Andor iXon LifeEMCCD camera, and ×100 Plan Apo and ×60 Plan Apo objectives.

**Co-localization analysis**. 3-dimensional Z-stack spinning disk confocal images of RBM20 and DDX6 were separately binarized using the Allen Institute Cell Segmenter (Centrin-2 pipeline for RBM20 and Dots pipeline for DDX6, exact segmentation parameters available upon request; https://www.allencell.org/segmenter.html). Segmented channels were then max projected and co-localization was measured using the image-based MeasureColocalization pipeline in CellProfiler (version 4.0.4)[74]. Measurement of co-localization reflects Pearson's correlation coefficient[75].

**RBM20 localization analysis**. To accurately quantify RBM20 localization, analysis of SIM images was required, as confocal techniques did not provide the required resolution (see Fig. 2). 3-dimensional Z-stack SIM images of RBM20 and DNA (Hoescht stain) were individually binarized using the Surfaces function in Imaris 9.7 (https://imaris.oxinst.com/). Two additional RBM20 masks were made using the nuclear mask as a fiducial marker. First, a "Nuclear RBM20 mask" was made by assigning all RBM20 within the nucleus a value of, "1". Second, a "Cytoplasmic RBM20 mask" was made by assigning all RBM20 outside the nuclear mask a value of, "0". Using these masks, the intensity value of RBM20 particles was measured in the nuclear and cytoplasmic compartments, and percent localization of nucleus vs. cytoplasm was calculated (e.g. percent in compartment divided by total signal). Statistical significance was calculated using a Student's $t$-test.

**Western blot**. Protein lysates were obtained with RIPA buffer containing 1× Halt protease inhibitor cocktail (Thermo Fisher Scientific). Following clarification of the lysate by centrifugation and assessment of protein concentration by BCA assay (Thermo Fisher Scientific), samples were diluted with 3× Blue Protein Loading Dye (New England Biolabs) and boiled for 5 min. Twenty micrograms of protein were separated via electrophoresis using 4– 20% Mini-PROTEAN TGX Precast Protein Gels (Bio-Rad). Proteins were transferred to PVDF membranes and blocked in TBS with 0.1% Tween-20 (TBST) and 5% Blotting Grade Buffer (BGB, Bio-Rad). The anti-RBM20 rabbit polyclonal primary antibody (Abcam, ab233147)) was diluted at 1:1000 in TBST 5% BGB and incubated overnight at 4 °C. Membranes were

washed three times in TBST for 10 min at room temperature, incubated for 1 h at room temperature with goat anti-rabbit HRP-conjugated secondary antibody, and washed three times in TBST for 10 min. The chemiluminescent reaction was initiated by incubation with SuperSignal West Pico Chemiluminescent Substrate (ThermoFisher), and images were acquired using a ChemiDoc Imaging System (Bio-Rad) in "high resolution" mode. Before re-probing for the housekeeping protein GAPDH (mouse monoclonal [6C5] diluted at 1:5000; Abcam #8245) according to the same protocol but using goat anti-mouse HRP-conjugated secondary antibody, membranes were treated with Restore Plus Western blot stripping buffer (ThermoFisher) for 15 min at room temperature, washed three times, and re-blocked. Band intensity was calculated by measuring intensity Fiji (Fiji Is Just ImageJ, https://imagej.net/Fiji) and normalized for background and GAPDH loading control[76].

**Reporting summary**. Further information on the research design is available in the Nature Research Reporting Summary linked to this article.

## Data availability

The sequencing datasets have been deposited in GEO (accession GSE175886) and Synapse (https://www.synapse.org/#!Synapse:syn2582579) as open-access data along with the processed data files. The previously published pig heart RNA-Seq was generously provided by Dr. Jay Schneider. All microscopy data generated in this study is available upon request. Source data, including genomics, PCR primers and complete blot images with standards are provided in the Source Data file. Source data are provided with this paper.

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

## Acknowledgements

This work was supported by generous research grants from the National Heart, Lung, and Blood Institute (U01 HL099997, P01 HL089707, R01 HL130533, F32 HL156361-01, R01 HL149734, R01 HL128362, R01 HL128368, R01 HL141570, R01 HL146868) to B.R.C., N.S., A.M.F., C.E.M., and N.J.S. National Institute of Diabetes and Digestive and Kidney (U54DK107979-05S1) to A.B. and C.E.M. National Science Foundation (NSF CMMI-1661730) to N.J.S. JSPS Grant-in-Aid for Young Scientists (A) [17H04993], NOVARTIS Research Grant, Mochida Memorial Foundation Research Grant, SENSHIN Medical Research Foundation Grant, Naito Foundation Research Grant, Uehara Memorial Foundation Research Grant, Uehara Memorial Foundation Research Fellowship, Gladstone-CIRM Fellowship to Y.M., and the A*STAR International Fellowship to K.K.B.T. G.W.Y. was partially supported by grants from the NIH (U54 HG007005 and R01 HG004659). Imaging was performed at the Gladstone Institutes' Histology and Light Microscopy Core, Garvey Imaging Core at University of Washington, Biological Imaging Facility at University of Washington and iPSC work was carried out in the Gladstone Institutes' Stem Cell core. We would like to thank Chi-Li Chiu for advice on quantitative image analysis, Samantha Bremner for advice and support on 3D-EHT experiments, and Deepak Srivastava and Shinya Yamanaka for their valuable advice on our data and the manuscript.

## Author contributions

A.M.F., Y.M., A.B., S.S., S.J.M., M.J.S., K.K.B.T., J.A.P-B., P.-L.S., G.W.Y., C.E.M., N.J.S., B.R.C., and N.S. designed the experiments. A.M.F. performed electrophysiological and contractile characterization of iPSC-CMs, and intracellular localization analysis of RBM20 with help from A.B. N.J.S. assisted with design and analysis of EHT experiments. Y.M., M.J.S., K.K.B.T., J.A.P.-B., A.H.C., S.J.M., T.D.N., C.R.R., P.P.L., A.T., and S.S. generated genome edited iPSC-CMs and conducted RNA-Seq and eCLIP analyses. A.K., K.C., and N.S. performed computational analyses. A.B., P.-L.S., G.W.Y., C.E.M., B.R.C., and N.S. supervised projects. A.M.F., Y.M., A.B., C.E.M, B.R.C., and N.S. wrote the manuscript with help from all authors.

## Competing interests

The authors declare the following competing interests: B.R.C. is a founder and has equity in Tenaya Therapeutics. N.J.S. is a scientific advisor to and has equity in Curi Bio, Inc. C.E.M. is a scientific founder and equity holder in Sana Biotechnology. G.W.Y. is co-founder, member of the Board of Directors, on the scientific advisory board, equity holder, and paid consultant for Locanabio and Eclipse BioInnovations. G.W.Y. is a visiting professor at the National University of Singapore. G.W.Y.'s interest(s) have been reviewed and approved by the University of California San Diego, in accordance with its competing interests policies. The other authors declare no competing interests.
