## [Peer Review File · Nature Communications]

Gain-of-function cardiomyopathic mutations in RBM20 rewire splicing regulation and re-distribute ribonucleoprotein granules within processing bodiesREVIEWERS' COMMENTS

Reviewer #1 (Remarks to the Author):

GENERAL COMMENTS

In this extensively revised manuscript by Fenix and colleagues, the authors examine the mechanism of pathogenesis of RBM20 in dilated cardiomyopathy. Particularly, the authors use the power of the iPSC model system and gene editing to explore how the R636S variant of RBM20 impacts the biology of cardiomyocytes. The authors' new manuscript not only describes splicing defects and differences that are unique to the R636S variant as they are not found in the simple RBM20 knockout, but also goes on to define the localization of mutant RBM20 in cytoplasmic ribonucleoprotein granules as well as to a lesser extent in the nucleus in contrast to wild-type RBM20 that exhibits essentially exclusive nuclear localization. There are a number of new insights as well as data confirming the original findings of misplicing. The authors' functional assays performing contractility analysis of the full complement of cell lines studied in engineered heart tissue improves significantly the functional data to complement the extensive molecular characterization. These contractility assays indeed are consistent with the heterozygous R636S constructs showing gain-of-function defects in contractility whereas the knockout does not clearly exhibit a difference from control. I am also impressed with the detailed time course of differentiation of the iPSCs over 60 days with extensive RNAseq data on gene expression as well as splicing analysis. These data are not only useful for defining the developmental time course for RBM20 expression and splicing but will be generally useful reference for the field as a whole. Overall, the authors put forward an important, data-rich manuscript honing in on the pathogenesis of RBM20-related cardiomyopathy with new insights regarding gain of function effects of the R636S variant mediated not only by unique splicing patterns for a multitude genes but also evidence regarding mislocalization of the variant contributing to disease. The manuscript opens many doors and questions regarding the misregulated pathways in the mutant iPSC-CMs that will require future study, but this body of work is an important step forward in advancing our understanding of this genetic cause of dilated cardiomyopathy.

SPECIFIC COMMENTS

Figure legend 1, "Dots represent individual 3D-EHTs from N = 2 for WT and N=3 for HTZ, HMZ, and KO 3D-EHTs independent biological replicates." There is an inconsistency here as there are more than 2 or 3 dots per bar graph.

Reviewer #2 (Remarks to the Author):

This extensively revised manuscript by Salomonis and colleagues is much improved and provides strong evidence for splicing-dependent and splicing-independent mechanisms for RBM20 in DCM pathogenesis. It brings together carefully considered set of experiments that support the intriguing conclusions that RS domain mutant of RBM20 has dominant-negative, gain-of-function properties.

I have no further concerns and feel the paper can be accepted for publication.

Reviewer #3 (Remarks to the Author):

This is a thoroughly revised manuscript. I would like to congratulate the authors to have invested a lot of time, resources, new data, and a more refined discussion to make this a compelling piece of work.

The authors have addressed all of my previous comments and concerns.

September 03, 2021

To:
Minju Ha, PhD
Senior Editor, Nature Communications

Dear Dr. Ha,

We thank the reviewers and editors for their time and repeated commitment to providing extremely helpful feedback and recommendations. We are excited for the opportunity to publish our research article at Nature Communication. Please find a revised manuscript and accompanying files that address the editorial and single reviewer comment.

In regard to the Reviewer 1, Specific Comment: “*Figure legend 1, “Dots represent individual 3D-EHTs from $N = 2$ for WT and $N=3$ for HTZ, HMZ, and KO 3D-EHTs independent biological replicates. “ There is an inconsistency here as there are more than 2 or 3 dots per bar graph.”*”, we can confirm that the dots represent individual 3D-EHT replicates from 2 independent biological differentiations for WT, and 3 independent biological differentiations for HTZ, HMZ, and KO genotypes.

We have worked to carefully address all of the editorial requests by:

- 1) Detailing specific versions of software
- 2) Clarifying the statistical tests and with associated revisions to figures as requested
- 3) Addition of specific Source Data files as per the editorial requests
- 4) Addition of requested technical experimental details regarding rigor
- 5) Renaming of figures and supplementary files as per Nature Communications requirements
- 6) Ensuring names of authors and affiliations are complete
- 7) Ensuring conflicts of interest have been detailed and vetted by all authors

Please let us know of any additional requests and we will work to provide the requested edits in a timely manner.

Sincerely yours,

Nathan Salomonis, PhD (on behalf of the co-authors)
Associate Professor,
Division of Bioinformatics, Department of Pediatrics,
Cincinnati Children's Hospital Medical Research Center